



**Transport and degradation of perchlorate in deep vadose zone: implications**
**from direct observations during bioremediation treatment**
**Ofer Dahan, Idan Katz, and Zeev Ronen**
Zuckerberg Institute for Water Research (ZIWR), The Blaustein Institutes for Desert
Research, Ben-Gurion University of the Negev, Israel
*Keywords:* Remediation, unsaturated zone, contaminant transport, perchlorate,
monitoring
**Abstract**
An in situ bioremediation experiment of a deep vadose zone (~40 m) contaminated
with a high concentration of perchlorate (>25,000 mg L$^{-1}$) was conducted through a
full-scale field operation. Favorable environmental conditions for microbiological
reduction of perchlorate were sought by infiltrating an electron donor-enriched water
solution using drip irrigation underlying an airtight sealing liner. A vadose-zone
monitoring system (VMS) was used for real-time tracking of the percolation process,
the penetration depth of dissolved organic carbon (DOC), and the variation in
perchlorate concentration across the entire soil depth. The experimental conditions for
each infiltration event were adjusted according to insight gained from data obtained
by the VMS in previous stages. Continuous monitoring of the vadose zone indicated
that in the top 13 m of the cross section, perchlorate concentration is dramatically
reduced from thousands of milligrams per liter to near-detection limits with a
concurrent increase in chloride concentration. Nevertheless, in the deeper parts of the



vadose zone (<17 m), perchlorate concentration increased, suggesting its mobilization
down through the cross section. Breakthrough of DOC and bromide at different
depths across the unsaturated zone showed limited migration capacity of biologically
consumable carbon and energy sources due to their enhanced biodegradation in the
upper soil layers. Nevertheless, the increased DOC concentration with concurrent
reduction in perchlorate and increase in the chloride-to-perchlorate ratio in the top 13
m indicate partial degradation of perchlorate in this zone. There was no evidence of
improved degradation conditions in the deeper parts where the initial concentrations
of perchlorate were significantly higher.

**1 Introduction**

In situ bioremediation of a contaminated unsaturated zone (also termed vadose zone)
depends mainly on the ability to control the hydrological, physical and chemical
conditions in the subsurface (Bombach et al., 2010; EPA, 2015; Höhener and Ponsin,
2014). Chemical and hydrological manipulations are primarily aimed at enhancing the
activity of specific indigenous degrading bacteria. The optimal conditions for specific
contaminants' degradation are usually determined in microcosm experiments, where
the preferred electron donor and acceptor for degradation can be controlled and
examined (Gal et al., 2008; Megharaj et al., 2011; Sagi-Ben Moshe et al., 2012). The
optimal degradation conditions, evaluated through laboratory experiments, usually
form the basis for selecting a strategy for in situ remediation in field-scale operations.
Nevertheless, implementation of desired biodegradation conditions in the deep vadose
zone through full-scale field setups requires control of the vadose zone
hydrogeochemical conditions. This is often achieved through either infiltration of


water enriched with electron donors or nutrients (EPA, 2004; Frankel and Owsianiak,
2005; Battey et al., 2007), or injection of a gaseous mixture capable of promoting
optimal biogeochemical conditions for microbial pollutant degradation (Evans and
Trute, 2006; Evans et al., 2011). Due to the complex nature of flow and transport
processes in the unsaturated zone, application of water with specific chemical
conditions near land surface does not necessarily result in promoting the desired
geochemical and hydraulic conditions in deeper parts of the vadose zone (Flury and
Wai, 2003; Jarvis, 2007; Allaire et al., 2009; Rimon et al., 2011a). Therefore, in the
vadose zone, and particularly in its deeper parts, a proper understanding of the
transport process is key to the success of in situ remediation operations (Dahan et al.,
2009; Rimon et al., 2011a; Baram et al., 2012a; Kurtzman et al., 2016).

Assessment of water percolation and solute transport in the vadose zone is

considered a major challenge in hydrological sciences. It is often characterized by
unstable flow that is highly sensitive to hydraulic, chemical and microbial conditions
(DiCarlo, 2007; Germann and al Hagrey, 2008; Dahan et al., 2009; Sher et al., 2012;
Hallett et al., 2013). Moreover, the chemical composition of the percolating water
[e.g., dissolved organic carbon (DOC), oxygen and nutrients] is subjected to frequent
changes due to natural hydroclimatic and biological cycles (Stumpp et al., 2012).
Accordingly, contaminant attenuation in the vadose zone is dependent on the complex
hydrological, chemical and biological states of the sediment. Continuous
measurements of the hydrological and chemical properties of the unsaturated zone
may be achieved with a vadose-zone monitoring system (VMS) (Dahan et al., 2009).
The VMS provides high-resolution measurements of variation in sediment water
content (Dahan et al., 2008; Rimon et al., 2007) and evolution of the pore water's





chemical composition across the unsaturated profile (Rimon et al., 2011a; Dahan et
al., 2014; Turkeltaub et al., 2014, 2016).
Perchlorate is an environmental pollutant that is often associated with the
explosives manufacturing industry (Roote, 2001; Urbansky, 2002; Trumpolt et al.,
2005). It is mostly produced, and consequently released to the environment as
ammonium perchlorate. Its high solubility (220 g $L^{-1}$) and stability in aerobic
environments makes it very mobile and persistent in the subsurface (Motzer, 2001;
Urbansky and Brown, 2003). Microbial reduction of perchlorate to harmless chloride
and oxygen in the unsaturated zone requires elevated water content, negative redox
potential, available electron donors and the presence of suitable indigenous bacteria
(Coates and Achenbach, 2004). In the vadose zone, natural attenuation and
biodegradation of perchlorate are considered very limited (Gal et al., 2009).
Nevertheless, studies have shown that perchlorate can be metabolized in unsaturated
soil whenever reducing conditions (<110 mV) (Attaway and Smith, 1993; Shrout and
Parkin, 2006) are achieved and an available electron donor is introduced (Tipton et al.,
2003; Frankel and Owsianiak, 2005; Nozawa-Inoue et al., 2005; Evans and Trute,
2006; Cai et al., 2010).
Here, the efficiency of a remediation operation of a perchlorate-contaminated
vadose zone was assessed using a VMS, which provided continuous information on
the chemical composition of the vadose-zone pore water. Promotion of perchlorate-
degrading conditions in the vadose zone was based on infiltration of water enriched
with ethanol (as a source of electron donor) from land surface. Real-time information
on the depth of the enriched water's propagation, along with variations in the
concentrations of perchlorate, chloride and bromide (applied as a tracer), was used to
assess transport and degradation of perchlorate across the unsaturated profile. Water-



and ethanol-application strategies were adjusted in each flow phase to obtain real-time
feedback on the chemical and hydrological state of the vadose zone.

**2 Study site**

The study area is located in the central part of the Israeli coastal plain, east of the city
of Ramat Hasharon. The site is a former unlined earthen pond that was used to store
industrial wastewater for several decades. A hydrogeological survey conducted in the
study area revealed substantial perchlorate contamination in the vadose zone and
groundwater under the pond area (Gal et al., 2008, 2009). It was concluded that
percolation of untreated wastewater from the ponds had crossed the 40m thick vadose
zone and created a large perchlorate pollution plume in the underlying phreatic aquifer
with concentrations exceeding 1,000 mg $L^{-1}$. In the vadose zone, however, the
investigation revealed extreme perchlorate pollution, reaching concentrations
exceeding 2,000 mg $kg^{-1}$ dry soil (equivalent to ~25,000 mg $L^{-1}$ in the sediment pore
water), along with high total salinity and chloride concentration exceeding 25,000 mg
$L^{-1}$. Because this area is under consideration for future urban development,
remediation of both the vadose zone and groundwater there is of major concern.
The stratigraphy of the area is characterized by Neogene and Pleistocene
sediments, mainly of sands and sandstones with interbedding of clay lenses
(Gvirtzmen, 2002). The vadose zone lithological profile at the site was assessed again
through a borehole that was drilled at the pilot site in 2012 (Table 1, Fig. 1). Most of
the profile is composed of yellow and red sand layers with low clay content (<5 %),
with interbeds of brown sand containing variable clay content of up to 11 %. A single
~1m thick clay layer (27.5 % clay content) was observed at a depth of 13.3 m. To



improve infiltration capacity in deep sections of the vadose zone during the
remediation experiment, a shallow clay layer with low permeability, known as
"nazaz" (Singer, 2007), was removed from a depth of 2.5–3 m by excavation. The
excavated area, 10 × 30 m, which was primarily assigned for the pilot infiltration
experiment, was backfilled with the sandy loam from the excavated site after removal
of the 0.5m thick nazaz layer. This layer is therefore presented in the profile as
disturbed soil.
***Table 1***. *Sedimentological Composition of the Vadose Zone at the Pilot Site*

| Depth (m) | Description | Clay Content (%) |
|---|---|---|
| 0–3 | Red sand (disturbed) | 7.5 |
| 3–5 | Red sand (Hamra) | 5 |
| 5–7 | Red-yellowish sand | 5 |
| 7–10 | Yellow sand | 5 |
| 10–13 | Brown sand | 5 |
| 13–14 | Dark brown clay | 27.5 |
| 14–17 | Red-brown clayish sand | 12.5 |
| 17–20 | Brown clayish sand | 3.75 |
| 20–27 | Yellow sand | 1 |
| 28–29 | Brown sand | 11.75 |
| 29–33 | Red-clayish sand (Hamra) | 3 |
| 33–41 | Yellow sand | 0 |


The climate in the area is characterized as subtropical Mediterranean with a hot and
dry summer from May to October and a colder wet winter from November to April.
The average air temperature on summer and winter days is 30 °C and 17 °C,
respectively. The average annual precipitation is 530 mm year[-1], mostly as rain
occurring mainly in four to seven rainy episodes during the winter season (IMS,

2011).



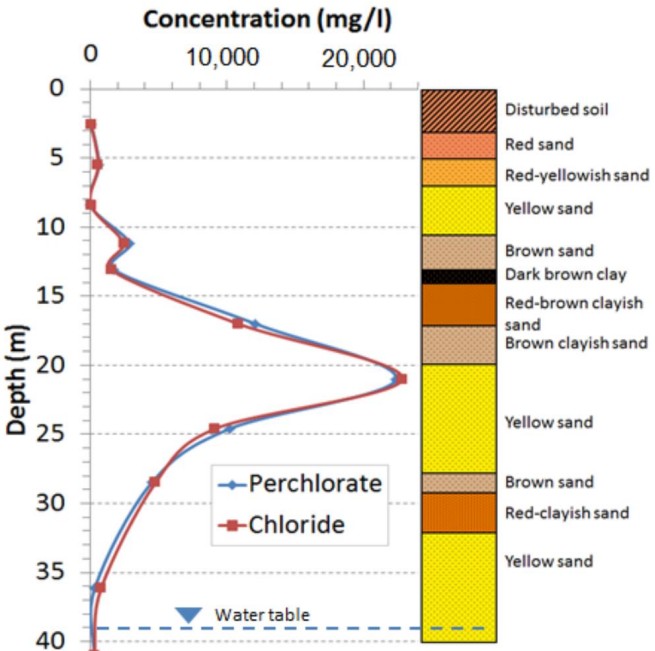


**Figure 1.** *Initial concentration profiles of chloride and perchlorate in the vadose zone*

*pore water under the former waste lagoon, along with the lithological profile.*

**3 Experimental setup**

**3.1 Vadose-zone monitoring setup**

Real-time characterization of flow and transport processes in the vadose zone, as well
as assessment of chemical transformation of the percolating water during the
remediation experiments were carried out with a VMS that was installed across the
entire unsaturated profile, from land surface to a depth of 37 m (Fig. 2). A detailed
description of the VMS, its structure, installation procedure and performance, can be
found in previous publications (Dahan et al., 2009; Rimon et al., 2011a) and in the





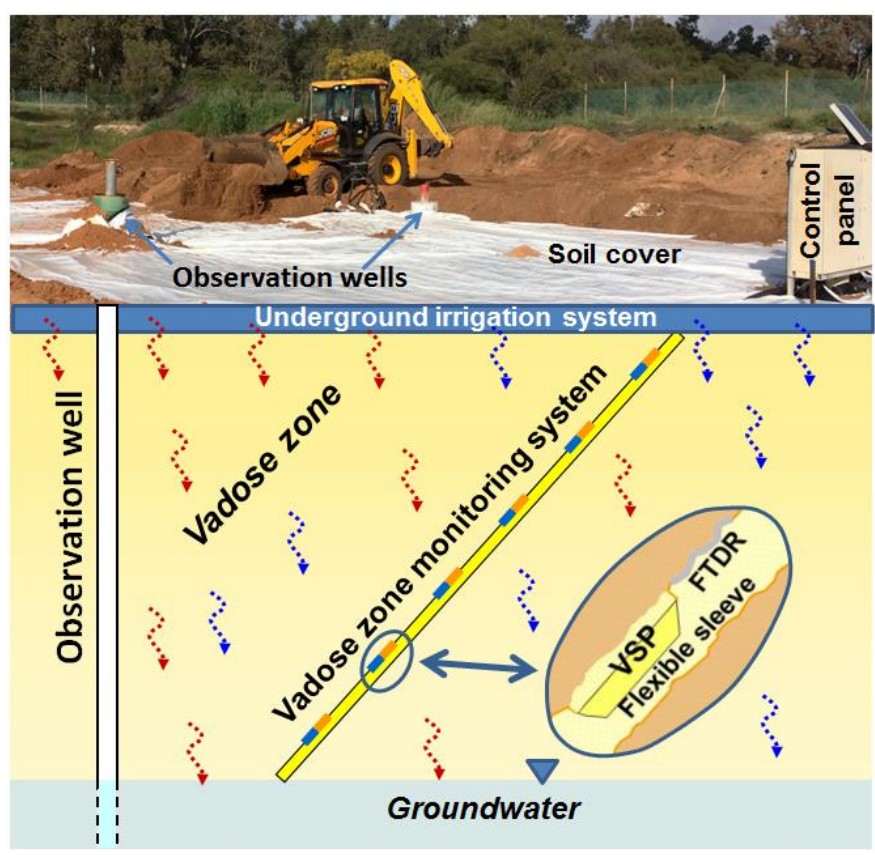


***Figure 2.*** *Schematic illustration of the vadose-zone monitoring system installed in the*

*vadose zone under the infiltration pilot site. In the picture above the vadose zone, the*

*irrigation system at the site is being covered.*

supplementary material. In particular, the VMS that was used at this site was
composed of a 44m long flexible polyurethane sleeve hosting 11 monitoring units
distributed along its length. Each monitoring unit included: (a) a flexible time-domain
reflectometer (FTDR) sensor for continuous measurement of variations in the
sediment water content (Dahan et al., 2008; Rimon et al., 2007), and (b) vadose-zone
sampling ports (VSPs), which enable frequent sampling of the vadose zone pore water
for chemical analysis (Baram et al., 2012a; Dahan et al., 2009; Rimon et al., 2011b;
Turkeltaub et al., 2016). The VMS flexible sleeve was installed in a 0.16m diameter



uncased borehole drilled slanted at a 55° angle (to the horizon) to a vertical depth of
37 m. In addition to the 11 monitoring units that were installed with the VMS, four
additional monitoring units were installed directly in the soil at depths of 0.5 m and
1.5 m. It should be noted that the slanted installation is preferred to ensure that
measurements carried out by each monitoring unit take place in separate undisturbed
sediment columns. In addition, the flexibility of the monitoring sleeve and its filling
with non-shrinking cement grout ensured complete sealing of the borehole void and
prevention of cross-contamination through preferential flow in the borehole.

**3.2 Field setup**

Water amended with ethanol as the electron donor for perchlorate-reducing bacteria
was infiltrated into the vadose zone through an area of 8 x 30 m at the pilot site using
a drip-irrigation system. Dripping lines with drippers having a nominal discharge rate
of 2.2 L h$^{-1}$ were set up in a 0.3 x 0.3 m spatial distribution to create fairly even water
distribution over the area. Accordingly, the total discharge rate of the irrigation
system was set to 5 m$^3$ h$^{-1}$, which is equivalent to an infiltration rate of 0.02 m h$^{-1}$. To
promote anaerobic conditions in the unsaturated zone, a polyethylene liner covered
with soil was placed over the dripper system after its installation. Ethanol was
selected as the electron donor and carbon substrate because it is a natural, soluble
compound that is commonly used by perchlorate-reducing bacteria (Bardiya and Bae,
2011). Moreover, it eliminates the increased soil salinity associated with other
common sources of electron donors such as acetate (Gal et al., 2008).

**3.3 Infiltration experiments**






Three infiltration experiments with variable amounts of water and ethanol were
implemented at the pilot site over a period of 7 months. To trace the percolating water
across the unsaturated zone, bromide (as KBr) was added to the infiltrating water at
the early stages of the experiment. The infiltration rates, as well as the concentrations
and application sequence were assigned for each experiment with insight gained from
the previous experiment (Table 2). Accordingly, information obtained by the VMS on
depth propagation of the ethanol and tracer and variations in perchlorate and chloride
concentrations across the unsaturated zone during and after each infiltration
experiment were used to adjust the infiltration procedure in the following stage.

*Table 2. Infiltration experiment conditions*

| Date | Water Volume $(m^3)$ | Equivalent Water Depth (mm) | Ethanol (l) | Bromide (Kg) |
|------|------|------|------|------|
| 8 Aug 2010 | 50 | 210 | 50 | 5 |
| 1 Sep 2010 | 100 | 420 | 50 | - |
| 27 Feb 2011 | 300 | 1250 | 200 | - |


The first experiment (8 Aug 2010) consisted of infiltration of 50 $m^3$ water
(equivalent to 210 mm) (Table 2). The first 6 $m^3$ were applied as untraced fresh water
with no ethanol to wet the topsoil. This wetting stage is essential to promoting deep
transport and preventing accumulation of tracers and ethanol in the low-flow zone
located on the margins of the dripper's influential zone. Following the initial wetting
phase, 0.4 $m^3$ of bromide tracer solution (as KBr) at a concentration of 12.5 g $L^{-1}$ was
applied, followed by 1 $m^3$ of water with 5 % ethanol. Immediately after the
application of the carbon and tracer solution, the rest of the water (42.6 $m^3$) was
applied to enhance transport of the ethanol and tracers to deeper parts of the vadose
zone.



After obtaining the results pertaining to the wetting process, as well as tracer

and ethanol migration in the vadose zone during the first infiltration experiment, a

second experiment was performed (1 Sep 2010). This experiment was conducted with

100 $m^3$ of water (equivalent to 420 mm). Here the first 7 $m^3$ of water was injected into

the topsoil as untraced fresh water, followed by 1 $m^3$ of water with 5 % ethanol, and

then the rest of the water dose (92 $m^3$). No tracers were used in this experiment. The

amount of water used after application of the ethanol was doubled to enhance

migration of the ethanol to deep sections of the unsaturated zone.

Results from the first two experiments indicated limited migration of tracer

and ethanol to deeper parts of the vadose zone. A third infiltration experiment was

therefore conducted 5 months later with increased discharge of 300 $m^3$ (equivalent to

1250 mm). This experiment started with 24 $m^3$ of untraced water followed by 0.4 $m^3$

concentrated (50 %) ethanol solution. Then, the rest of the water (275.6 $m^3$) was used

to push the ethanol down into the vadose zone. The large quantity of water applied

after the concentrated ethanol solution was designed to enhance quick migration of

the ethanol to deep parts of the vadose zone while minimizing its biodegradation in

the upper soil layers.

**3.4 Analytical procedure**

Perchlorate was analyzed with a perchlorate ion-selective electrode (ISE; Laboratory

Perchlorate Ion Electrode, Cole-Parmer, USA). All samples measured with the ISE

were adjusted by dilution to a concentration range of 10–100 mg $L^{-1}$. Duplicates were

frequently analyzed by injecting 25 μL sample into a Thermo Scientific™ Dionex™

ion chromatography system (ICS 5000) equipped with  Ion Pac AS19 column





(detection limit of $\pm 0.01$ mg L$^{-1}$). Because results from the two methods were not
significantly different, most of the data reported here are from the perchlorate
electrode with a detection limit of 1 ppm. Bromide and chloride were analyzed by ion
chromatography with a detection limit of 30 ppb (Gal et al. 2008). Total organic
carbon (TOC) was analyzed to examine the success of delivering carbon to the vadose
zone. Because porewater samples from the vadose zone are obtained through the VSP,
which uses a porous ceramic interface (pore size < 2 μm), TOC values reflect DOC.
TOC was analyzed through a combustion TOC analyzer (Teledyne Tekmar, Apollo
9000) with a detection limit of 2 ppm. Ethanol concentration in the vadose zone pore
water was analyzed in a gas chromatograph (Varian, CP3800). Water samples (1.5
μL) were injected by autosampler. The FID and injector temperatures were set to 270
and 250 °C, respectively. The GC oven temperature was first held at 50 °C for 1 min,
increased to 220 °C at a rate of 25 °C min$^{-1}$, and then held for 4 min. The separation
was performed by Stabilwax® capillary column (60 m, 0.32 mm, 0.25 μm, Restek
Corporation, USA); helium was used as the carrier gas (1 mL min$^{-1}$). For
quantification, five external standards were used.

**4 Results and discussion**

All of the data obtained by the VMS are presented here as variations in measured
parameters with depth, as commonly done to describe depth profiles. However, to
ensure measurements under undisturbed vertical profiles, the VMS was installed in a
slanted orientation (Fig. 2 and supplementary material). Thus, each monitoring unit
faces an undisturbed profile that is shifted horizontally and vertically from the other
units. Accordingly, although the data are presented as depth profiles, they should be

                                             *Dahan*



regarded as individual points distributed across the 3D space of the vadose zone
(Dahan et al., 2007; Rimon et al., 2011a).

**4.1 Water percolation**

Temporal variations in the vadose zone water content provide a direct indication of
percolation processes in the vadose zone (Rimon et al., 2007; Dahan et al., 2008;
Turkeltaub et al., 2015). Each infiltration experiment launched a wetting wave that
propagated sequentially through the unsaturated zone (Fig. 3). Down-migration of the
wetting wave was expressed as a quick rise in water content followed by a recession
caused by water redistribution and drainage. Referring the wetting sequence in the
vadose zone to the infiltration events on land surface enabled a direct calculation of
the flow velocity across the unsaturated zone (Rimon et al., 2007; Dahan et al., 2008).
All three infiltration experiments produced wetting fronts that moved down the
vadose zone at a velocity of ~0.18 m h$^{-1}$, even though the water volumes that were
used in each experiment were significantly different (50, 100 and 300 m$^3$).
Observations of regulated flow velocities at constant rates across the vadose zone
under variable surface hydraulic conditions have also been reported in other studies
(Dahan et al., 2008; Amiaz et al., 2011; Rimon et al., 2011a).
The high salinity of the deeper parts of the vadose zone (>13 m) (Fig. 1) limits
the reliability of the TDR technology for measuring water content at those depths
(Nadler et al., 1999). Therefore, variation in water content, as an indication of deep
percolation, is presented here only down to a depth of 11.2 m, where the salinity was
low enough to achieve reliable moisture measurements with the FTDR sensors.
Nevertheless, indications of deep percolation in the deeper layers (>13 m) are further




discussed through the variation in chemical composition of the percolating water
across the entire thickness of the unsaturated zone (40 m).

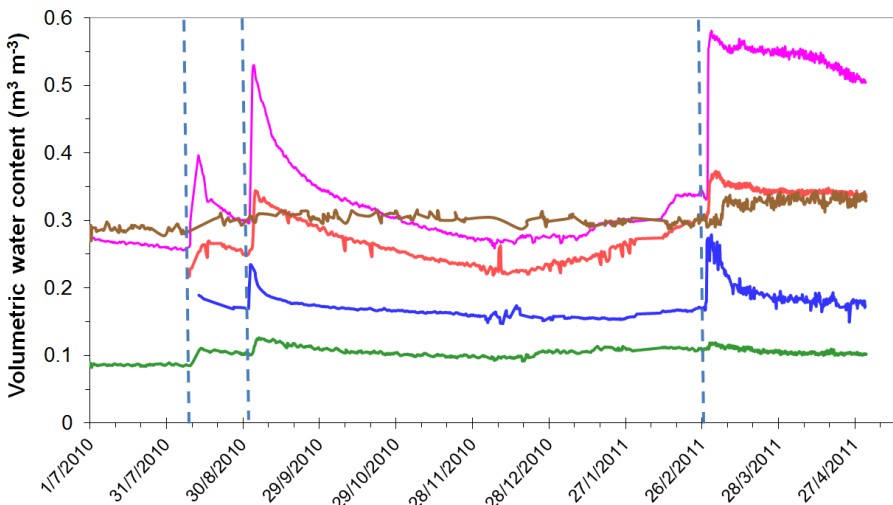


***Figure 3.*** *Temporal variations in sediment water content in the top 13 m of the vadose*
*zone during the infiltration experiments. Dates are given as day/month/year.*

**4.2 Perchlorate transformation and mobilization**

Initial analysis of porewater samples from the vadose zone, prior to initiation of the
infiltration experiments, revealed very high concentrations of perchlorate and
chloride, both reaching maximum values of ~22,500 mg L$^{-1}$ (Fig. 1), and total
dissolved solids (TDS) of 43,000 mg L$^{-1}$, at a depth of 21 m. Note that at this stage,
the concentrations of perchlorate and chloride are nearly identical throughout the
entire profile. These high concentrations, sampled by the VMS, are in accordance
with concentration profiles obtained previously in extracts of sediment samples (Gal
et al., 2009).





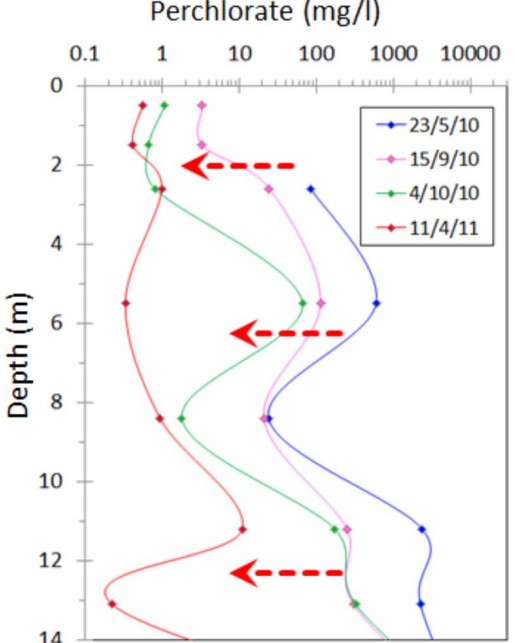


**Figure 4.** *Perchlorate concentration profile across the top 13 m of the vadose zone*


*under the pilot site during the infiltration experiments. The profiles emphasize the*


*gradual decrease in perchlorate concentration with time. Dates are given as*


*day/month/year.*


Frequent sampling of the vadose zone pore water showed dynamic variations
in perchlorate concentration during the percolation experiments. In the upper section
of the vadose zone (0–13 m), perchlorate concentrations decreased dramatically, from
as high as 9000 mg $L^{-1}$ to below detection levels (Fig. 4). Such a reduction in
concentration in a relatively thick portion of the vadose zone (13 m) over the short
period of 10 months is clearly desirable and may even be considered a great success.
Nevertheless, closer inspection of the variations in perchlorate concentration in deep
parts of the vadose zone (17–40 m) showed an increase at most of the measurement
points (Fig. 5). Perchlorate concentration rose from 12,700 mg $L^{-1}$ to 27,400 mg $L^{-1}$ at




a depth of 17 m during the same period. A similar increase in concentration was also
found in deeper parts of the cross section at depths of 25, 28, and 36 m. Note that
during this period, an increase in perchlorate concentration was even observed in the
groundwater (represented at a depth of 41 m in Fig. 5). Obviously, the mixed trend in
variations of perchlorate concentration implies that transformation and mobilization
processes take place simultaneously. As such, the conditions for both biodegradation
and  mobilization  should  be  examined  along  with  the  variation  in  perchlorate
concentration.

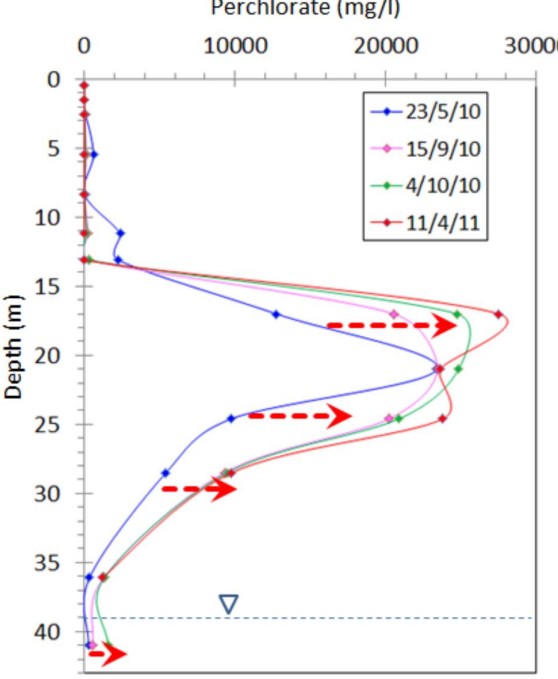


**Figure 5.** Perchlorate concentration profile across the entire vadose zone and top
groundwater under the pilot site during the infiltration experiments. The profiles
emphasize the gradual increase in perchlorate concentration with time. Dates are
given as day/month/year.



### 4.3 Electron donor availability

Available organic carbon as an electron donor is crucial for perchlorate reduction. To increase the concentration of DOC in the vadose zone, ethanol was mixed with the percolating water during the early stage of each infiltration experiment. Analysis of ethanol and DOC in the water samples from the vadose zone throughout the experiment revealed a similar concentration pattern (1 g ethanol = 2 g DOC). Therefore, we assume that the variation in DOC was due to transport of ethanol or ethanol-degradation products with the percolating water.

During the first infiltration experiment, an increase in DOC above background levels was observed only at shallow depths, down to 1.5 m (Fig. 6). No signs of increasing DOC were observed in the deeper parts of the cross section at this stage. Twenty-three days later, before initiation of the second infiltration experiment, DOC values had dropped back down to background levels. This implies that the ethanol was microbiologically consumed in the soil before it could be leached further down.

As a result of the limited transport of ethanol in the first infiltration experiment, a second experiment was conducted with the same mass and concentration of ethanol. However, it was flushed with double the amount of water to promote its quicker migration to deeper layers (Fig. 4). In this experiment, no signs of increasing DOC were observed at any depth. On the contrary, DOC level decreased to values below background levels (Fig. 6). Obviously, the rate of ethanol metabolism in the soil increased following the first experiment, where both water content of the sediment and substrate required for efficient microbial activity increased. As a result, ethanol-degradation efficiency in the topsoil (<0.5 m) was significantly enhanced.



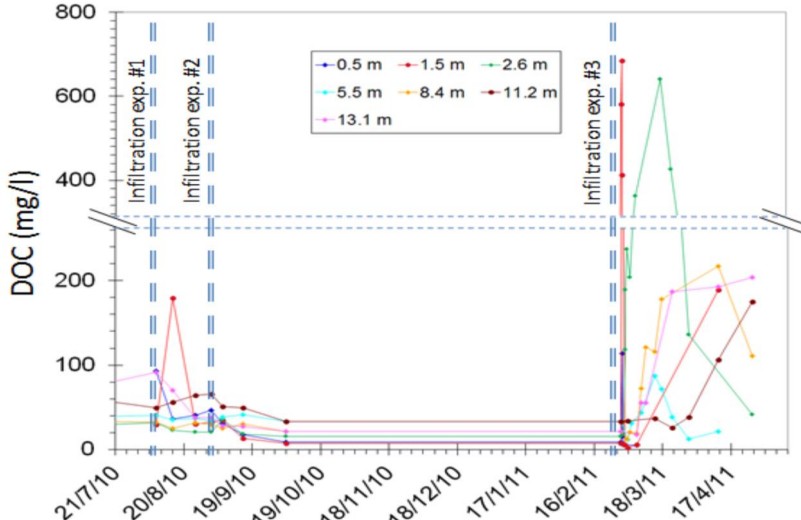

**Figure 6.** *Variations in dissolved organic carbon (DOC) across the top 13 m of the*

*vadose zone following infiltration of water enriched with ethanol. Dates are given as*

*day/month/year.*

To overcome the limitation of electron donor delivery through the shallow

soils, a third infiltration experiment was designed. In this experiment, the ethanol was

injected in a 0.4m$^3$ high-concentration (50 %) pulse followed by a large volume of

water. Application of ethanol at a very high concentration was aimed at suppressing

its biological degradation in the shallow soil. The ethanol pulse was introduced after

application of 24 m$^3$, the latter to provide high initial wetting conditions under the

ethanol front. Then the ethanol slug was pushed down with 276 m$^3$ of water. At this

stage of the study, which was conducted 6 months after the previous one, a substantial

increase in DOC was observed in the entire top 13 m of the cross section (Fig. 6).

Obviously, an increase in DOC serving as electron donor is an essential prerequisite

for perchlorate degradation. Apparently, application of ethanol at a high

concentration, which inhibited its degradation in the upper layer, succeeded to drive




the ethanol all the way down to 13 m, just above the clay layer. Nevertheless, no signs
of DOC increase were observed below 13 m.

**4.4 Transport and degradation**


The mechanism controlling down-propagation of a non-conservative substance such
as ethanol may be elucidated by looking at the migration pattern of a conservative
tracer such as bromide. Bromide was injected with the percolating water in the early
stages of the first infiltration experiment. Results on bromide migration are presented
here only for the top 13 m, where the background concentrations prior to the initiation
of the infiltration experiment were below detection limits. Concentration profiles
during the infiltration experiments clearly demonstrated sequential progress of the
percolating water across the top 13 m of the unsaturated zone (Fig. 7). Mass balance
calculation of bromide on the basis of the concentration profiles (Fig. 7) and sediment
water content (Fig. 3) on various dates after the infiltration experiment resulted in
high recovery rates of 85–127 %. A comparison of the transport patterns of bromide
and DOC confirmed that biodegradable material such as ethanol is rapidly consumed
in the vadose zone.
An increase in chloride concentration in the vadose zone is usually attributed
to evaporation processes near land surface, a mechanism that is unlikely to occur in
this particular setup where the surface is isolated from the atmosphere. Accordingly,
variations in chloride concentration across the vadose zone may be attributed to
chloride mobilization with the percolating water and perchlorate reduction. Therefore,
degradation of perchlorate is expected to result in an increase in chloride mass.



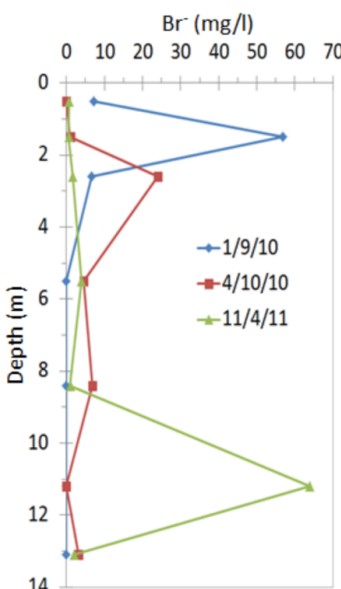


**Figure 7.** *Variations in bromide concentration profile across the top 13 m of the*

*vadose zone during the infiltration experiments. Dates are given as day/month/year.*

Prior to the infiltration experiments, chloride-to-perchlorate ratios in the
vadose zone were very similar, exhibiting nearly identical profiles (Fig. 1) with
equivalent concentration proportions of 2.4–5.5 (Fig. 8). Following the infiltration
experiment, a significant increase in ionic ratios was observed in the top 13 m, while
in the rest of the profile—from a depth of 17 m to the water table, the concentration
ratio of chloride to perchlorate remained relatively unchanged. Obviously, since both
perchlorate and chloride are very soluble and mobile, infiltration water with a low
concentration of chloride ($\sim$100 mg L$^{-1}$) and zero perchlorate is also expected to result
in an increased chloride-to-perchlorate ratio, even if no perchlorate degradation takes
place. Since both chloride and perchlorate are very mobile and easily displaced with
the percolating water, quantification of the perchlorate-degradation rate with respect
to its down-leaching is not straightforward.

                                                                           Dahan



**Figure 8.** *Chloride-to-perchlorate equivalent concentration ratio profiles before and after the infiltration experiments. Dates are given as day/month/year.*

**5 Conclusions**

The infiltration experiments were primarily aimed at improving the environmental conditions for perchlorate-reducing bacteria across the vadose zone. This included an increase in water content along the soil profile and amendment of the electron donor. The results, which were based on continuous monitoring of the entire vadose zone, exhibited notable variation in the concentrations of perchlorate, DOC and other solutes in the unsaturated zone. Increased concentrations of DOC with a concurrent reduction in perchlorate concentration (from thousands to a few milligrams per liter) and increased chloride-to-perchlorate ratio (from ~2.5 to ~300) in the upper 13 m



indicated that perchlorate is partially reduced in this part of the vadose zone. On the
other hand, no evidence of improved reducing conditions was observed in the deeper
parts, where the initial concentrations of perchlorate were significantly higher.
Nevertheless, since assessment of redox conditions in deep vadose zone is not yet
feasible, we can only rely on variations in the chemical composition to assess the
existence of degradative conditions.

The limited ability to deliver a soluble electron donor across a

microbiologically reactive medium, such as topsoil, is a major limiting factor for
remediation of the deep vadose zone through gravitational percolation of enriched
solution. Note that temporal variations in the concentrations of perchlorate, as well as
other solutes, in the deep parts of the vadose zone, i.e., under the clay layer at 14 m,
indicate that the clay layer does not play any role in limiting infiltration capacity.
Similar observations on the role of clay layers in infiltration in the unsaturated zone
have been reported in previous publications (Rimon et al., 2007; Baram et al., 2012a,
2012b).

The attempts to leach the ethanol down into the vadose zone with large

quantities of water inevitably drove down-leaching and displacement of the dissolved
solutes, including perchlorate. Although there were indications of partial degradation
of perchlorate in the upper part of the vadose zone, its downward displacement toward
the water table was evident from the sequential increase in perchlorate concentration
with depth (Fig. 5). It seems that the entire column of perchlorate mass was pushed
down by the percolating water toward the water table, which also resulted in an
increased concentration of perchlorate in the observation well, which was located
under the infiltration zone.



The study demonstrates that application of vadose-zone monitoring
technology during a remediation operation provides real-time information on the
chemical and hydrological state of the subsurface. Linking the temporal variation in
the chemical composition of the vadose zone pore water, sediment saturation degree
and flow velocities are vital for efficient management of remediation operations.

***Author contribution:*** *Ofer Dahan (PI, Vadose zone hydrology) design of the*
*experimental and monitoring setup. Idan Katz (MSc student) conducted the field*
*experiment and laboratory analysis. Zeev Ronen (PI, Microbiology) design the bio*
*treatment setup. Data analysis and manuscript preparation - all coauthors.*

***Competing interest:*** *The authors declare that they have no conflict of interest.*

***Acknowledgments***. The authors wish to express their appreciation Israeli Water
Authority for project funding.






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
