# Peer review of "Transport and degradation of perchlorate in deep vadose zone: implications"

_Hydrology and Earth System Sciences, 2016_

## Referee Comment (RC1) · Anonymous Referee #1 · 8 Feb 2017

General comments

The authors present results of an experimental approach for remediating a polluted site. They made use of microbial processes in the deeper parts of the soil for reducing perchlorate pollution within the vadose zone triggered by the application of ethanol as electron-donor. The success of the infiltration and of the perchlorate reduction was monitored by means of a vadose zone monitoring systems. This system allows for a continuous observation of soil moisture and the sampling of soil solution in different depths. The experiment was carried out in three steps with increasing quantities of water and ethanol. First results showed that the initially applied water and ethanol quantities were not sufficient for a deep infiltration into the polluted zone of the soil.

The results indicate that both infiltration and perchlorate reduction could be triggered at least after the third application.

The presented topic is of relevance for many sites worldwide, polluted with different chemicals which can be deactivated by microbial processes. The specific challenge of this approach was the location of the pollution within a deep vadose zone with complicated water flow conditions. In addition, the chosen approach was based on natural seepage and not on a forced washing of the soil by infiltration wells. Thus, the topic is scientifically interesting and seems to be relevant for publication in HESS.

The paper is generally well-written and of good quality with a language which is concise and well understandable. The experimental setup as well as the results is presented clearly. The conclusions are generally comprehensible and objective.

However, with regard to the order and content of the subsections some improvement for a better understandability could be made. Some context would be easier to understand if the order of subsections would be rearranged. For example: Section 4.3 explains why the different treatments for the experiments were chosen, because the infiltration depth was not sufficient in the beginning and the concentration of ethanol was too low during the first experiment. It would be good to have this information already in the beginning before the results of perchlorate transformation are shown and discussed. The same is true for the presentation of bromide tracer behavior (in the beginning of section 4.4) which again explains the experimental setup. Please think about a change in the order of these parts of the results chapter.

Specific comments

p. 5, l. 111: You state that perchlorate is slowly leached into the groundwater. Can you describe the behavior of this pollutant in the saturated zone? Is it reduced or only transported by groundwater flows?

p. 6, l. 147: What is the effect of these climatic conditions? Is the perchlorate only

transported during the winter season and probably rises again during summer due to capillary action?

p. 11, l. 229: Please explain why no tracer was used in the second and third application.

p. 12, l. 272 Can you exclude lateral fluxes of seepage water?

p. 15, l. 326: Is the described successful reduction of perchlorate concentration the result of transport or reduction processes? Would it be a success if perchlorate is mainly transported by seepage water into deeper parts of the soil?

p. 16, l. 333: You mention mixed trends for both transformation and mobilization processes. Could you explain this conclusion more in detail?

p. 17, l. 350: Probably the relation between ethanol concentration and DOC could be shown by means of a figure and a regression curve?

p. 21, fig. 8: Is the red graph an average for data of the period 1/3-11/4 2015 (1.5 months)?

p. 22, l. 459: You end up with the conclusion that the entire column of perchlorate was pushed downwards by the infiltrating water. Thus, the problem is mainly shifted to the groundwater. Could you discuss the overall success of the presented remediation experiment against this background?

Technical corrections

References: Bauterse et al (2000) and Stumpp et al. (2009) are not mentioned in the text

Fig. 3: the legend is missing

Fig. 4/5: explain the meaning of the red arrows

[Figure]

---

## Referee Comment (RC2) · M Vanclooster (Referee) · 27 Feb 2017

General comments This paper describes an interesting large-scale vadose zone infiltration experiment, aiming to drive and understand the transport of a perchlorate pollution plume in a 40 m thick vadose zone. In the experiment, 3 ethanol-enriched water infiltration pulses are applied to a limited field site by means of a regularly distributed drip irrigation system. Water content, perchlorate content, chloride content and DOC is monitored at different depths (11 stations) by means of a VMS system designed in a previous work. Results of the monitoring allow elucidating depth-time patterns of water, perchlorate, chloride, TDS and DOC which conceptually are related to perchlorate transport and degradation. The conceptual results are suggested to be used in the

design of the perchlorate remediation process.

The paper enters in the scope of the journal. The paper is original since very few large scale experiments are reported allowing to monitor the fate of pollution plumes throughout the vadose zone during the pollution remediation process. Yet, although the experimental work is impressive, the study lacks some details and conceptual founding for accepting this study in this stage as a full publication in HESS. The major concerns are i) the absence of any quantitative modelling of the water transport and/or the perchlorate pollution plume during the infiltration experiment; ii) the absence of any uncertainty assessment. Hypothesis related to the fate of the perchlorate plume are indeed subjected to the hypothesis of mass conservation and representativity of the singular sampling. These strong hypotheses can only be considered acceptable in the present case if the experimental results are compared with some quantitative modelling that are built on mass conservation principles ( using e.g. a numerical water and solute transport, or NAPL/DNAPL transport model). As long as this numerical modelling is not added to the paper, the results remain too much speculative. I, therefore, recommend rejection of the paper in the current form, but with a strong recommendation to resubmit the paper in which a numerical modelling component has been added.

Specific comments

Line 103. Study site. Can the origin of perchlorate in the study site be identified?

Line 121. Heterogeneity in sedimentary vadose zone formations is omnipresent. Hence, how reliable is the single borehole to assess the lithology of the study site. Is the information of the borehole consistent with information obtained from the boreholes in the vicinity of the sampling point?

Line 152. The high suspected correlation between chloride and perchlorate concentrations demonstrates that there is some natural attenuation. This is in contrast with the statement in the literature review (line 86).

[Figure]

Line 198. Explain more in detail how ethanol can eliminate increased salinity.

Line 214. Specify for each infiltration pulse how much time was needed to apply the water/tracer/ethanol (hence the application rates). Also, add an estimate of the saturated hydraulic conductivity of the different layers to demonstrate that the infiltration rates stayed sufficiently below the ponding infiltration rate.

Line 250. Significant at which statistical level?

Line 287. Specify exactly how the wetting front velocities are determined. We are definitely in strong transient flow conditions. Hence the wetting front velocities will vary dynamically in time.

Line 290. Be more rigorous and more specific with respect to 'flow velocities'. How are these "flow velocities" defined in a heterogeneous and time dynamic flow system? (Cf. a major concern on the need to confront such statements with those from a quantitative numerical model).

Line 302. Legend incomplete. What are the different coloured curves? Where are the results of the 11 sampling units? Quid results of the control units in the top layer (0,5 and 1.3 m depths)?

Line 302. Explain more in detail the observed curves. E.g. what happens with the TDR probe at the top (I suppose) during the third infiltration event? The drainage curve looks completely different. So what happened?

Line 356. This statement can't be supported. This can only be concluded if mass conservation is checked. You can have lateral flow dissipation in such system. Only, a comparison of the results with the results of a numerical mass conservative model can support such conclusions.

Line 400-402. Show this in an explicit way.

Line 426. Confusing legend. 1/3 -11/4 2011. Specify which data at which date exactly.

Line 451. There are other studies showing that the clay layers will have considerable impact on the vadose zone dispersion (See e.g. Javaux M. and M. Vanclooster, 2004. In situ long-term chloride transport through a layered, non-saturated subsoil.1. Data set, interpolation methodology and results. Vadose zone journal 3 : 1331-1339.).

Line 461. This has not been shown in the paper.

————————————————

---

## Author Comment (AC1) · 19 Mar 2017

Reply to reviewer #1 comments on the manuscript:

Transport and degradation of perchlorate in deep vadose zone: implications from direct observations during bioremediation treatment

We thank the reviewer for his constructive review and intend to address all of his comments. We would like to state that we are specifically encouraged by his statement "The presented topic is of relevance for many sites worldwide, polluted with different chemicals which can be deactivated by microbial processes. The specific challenge of this approach was the location of the pollution within a deep vadose zone with compli-

cated water flow conditions". In our view this is the main essence of this manuscript.

General comments

Comment: Some context would be easier to understand if the order of subsections would be rearranged. For example: Section 4.3 explains why the different treatments for the experiments were chosen, because the infiltration depth was not sufficient in the beginning and the concentration of ethanol was too low during the first experiment. It would be good to have this information already in the beginning before the results of perchlorate transformation are shown and discussed. The same is true for the presentation of bromide tracer behavior (in the beginning of section 4.4) which again explains the experimental setup.

Reply: We accept the comment. A section describing the overall structure of all three experiments will be added to the beginning of the result chapter. It will present the rationale behind all experiments and gives an overview of the measurements before detailed description of the various components. Specific comments

Comment: p. 5, l. 111: You state that perchlorate is slowly leached into the groundwater. Can you describe the behavior of this pollutant in the saturated zone? Is it reduced or only transported by groundwater flows?

Reply: Perchlorate is well known to be fairly stable in groundwater. Its natural degradation is very limited and it is highly mobile. This has been presented in several publications (See for example a review paper by Bardiya et al. 2011, a chapter in a book Coates JD, Gu B. 2006, and perchlorate mobilization in this particular site Gal et al. 2009 (all of which are cited in this manuscript). The possibility of perchlorate reduction is depend in the groundwater redox conditions. We had reported in the past that groundwater is aerobic and thus natural degradation of perchlorate is not expected (Bernstein et al., 2010). Since our manuscript focus on the vadose zone where the hydro-chemical and biological conditions are substantially different from those occurring in groundwater we rather to focus on the vadose zone and not elaborate on the

saturated part beyond the limited citations in the introduction chapter.

(Bernstein, A., Adar, E., Ronen, Z., Lowag, H., Stichler, W., & Meckenstock, R. U. (2010). Quantifying RDX biodegradation in groundwater using $\delta$ 15 N isotope analysis. Journal of contaminant hydrology, 111(1), 25-35.)

Comment: p. 6, l. 147: What is the effect of these climatic conditions? Is the perchlorate only transported during the winter season and probably rises again during summer due to capillary action? Reply: The vadose zone is very thick ($\sim$40 m) and mostly sandy. As such capillary action is not relevant and will not impact more than the bottom $\sim$1 m. The experimental area has been covered with a sealing liner to prevent air penetration and to promote reducing conditions in the vadose zone. As such the only source of water to the subsurface in this period is the water injected to the soil with the drip irrigation system under the surface cover. Accordingly the consequence of rain water infiltration is eliminated. In addition in such thick vadose zone even seasonal temperature fluctuations are limited to the upper 2 m (Rimon et al. 2011b, cited in the manuscript). As such we believe that the climate has only limited impact on the conditions in the subsurface.

Comment: p. 11, l. 229: Please explain why no tracer was used in the second and third application. Replay: A single slug of tracer was used in in the beginning of the first experiment. It was designed to enable tracing of the wetting front that was introduced to the subsurface during the experiment. Application the tracer in the following experiment would have result in smearing the identity of the front and masking our capability to trace the moving water. In well-defined medium such as column experiment it is possible to differ between tracers applied in different stages. Yet we tend to believe that in natural heterogeneous system where water flow may be subjected to multi flow trajectories that may be activated and deactivated according to the hydraulic condition (see Dahan et al. 2009), application of the tracer in the following experiments would be a disadvantage.

Comment: p. 12, l. 272 Can you exclude lateral fluxes of seepage water?

Reply: We cannot absolutely exclude local limited of lateral fluxes. Nevertheless, creation of lateral flow in the unsaturated zone require, by definition, generation of saturated conditions that will create positive pressure which could overcame gravitational drainage. Up to date the vadose zone monitoring system has been installed in dozens of sites with different geological and hydrological conditions (See for example Dahan et al., 2007, Dahan et al. 2008, Rimon et al 2007, 2011a, 2011b, Amiaz et al 2012 and others). In none of these sites we found evidences for creation of saturation conditions and thus creation of lateral flow in the vadose zone, even though some of the sites were under flooded conditions of high water head (Dahan et al 2007, 2008), some with geological formations which are composed of clay interbeds that could potential create some kind of hydrological barrier and lateral flow. Since we did not find any indication for lateral flow in any of the other studies where water flow in the vadose zone was monitored we tend to believe that in this particular site lateral flow, if any, was very limited. In this discussion we ignored lateral small scale capillary flow and lateral flow in purged aquifers. Both are not relevant to this site.

Comment: p. 15, l. 326: Is the described successful reduction of perchlorate concentration the result of transport or reduction processes? Would it be a success if perchlorate is mainly transported by seepage water into deeper parts of the soil?

Comment: p. 16, l. 333: You mention mixed trends for both transformation and mobilization processes. Could you explain this conclusion more in detail?

Reply to the two comment above (p.15 and p.16): This comment emphasize the greatest challenge we faced in this project. Can we absolutely state that the reduction in perchlorate concentration that we have observed in the upper parts of the unsaturated zone are the result of bio-degradation or simple down leaching with the percolating water. Moreover, we have to investigate this question in light of the fact that the concentration of perchlorate in some deep section only increased during the infiltration

experiments. Throughout the paper we have discussed the potential degradation versus leaching from different prespective. In section 4.3 we have analyzed the potential degradation of perchlorate to the availability of electron donor. Obviously under absence of available electron donor no perchlorate degradation will take place. Though we managed to introduce electron donor into the vadose zone it was limited to the top 13 m. only there we found some reduction in perchlorate. In the rest of the profile we found no increase in available electron donor and in fact we also found no reduction perchlorate concentration. On the contrary in some places the concentration only increased which is an obvious indication to perchlorate mobilization with the percolating water. Further down in the manuscript in section 4.4 we discussed the potential degradation of perchlorate versus its transport through a comparison of the ethanol migration, which was consumed, versus the tracer, Br. Here we also compared the reduction in perchlorate with the variations in concentration of the degradation by-product chloride across the unsaturated zone and found a pronounced increase in Cl/Perchlorate only in the zones where we found available electron donor. All of these indicators provided hints to the question on the degradation vs leaching.

In the second part of the first comment the reviewer ask if "it be a success if perchlorate is mainly transported by seepage water into deeper parts of the soil". This is a very important question that is the subject of several studies we are conducting now (See Avishai et al 2016. Journal of Hazardous Materials). Since we found that achieving "efficient" degrading conditions in deep vadose zone is limited and we found that perchlorate mobilization in the unsaturated zone is very high we are testing the possibility to leach the pollution down to the groundwater where it can be retrieved back for treatment on land surface.

Comment: p. 17, l. 350: Probably the relation between ethanol concentration and DOC could be shown by means of a figure and a regression curve?

Reply: As mentioned in p.17 lines 346-352, we found high correlation between ethanol and DOC. Even though ethanol is mineralized by perchlorate reducing bacteria, it may

degraded first to acetate that also serve as energy for the degrading bacteria thus, DOC provide better picture on the availability of electron donor in the soil pore water. Since it is all presented in the manuscript text we believe that adding this information in a figure is somewhat not necessary. Figure 1 below display ethanol vs DOC in all water samples were both ethanol and DOC were measured Comment: p. 21, fig. 8: Is the red graph an average for data of the period 1/3-11/4 2015 (1.5 months)?

Reply: The red graph is a combination of data obtained from two consequent sampling data. Due to a technical problem that was resulted in luck of samples from one of the dates it was necessary to integrate data from these two consequent dates.

Comment: p. 22, l. 459: You end up with the conclusion that the entire column of perchlorate was pushed downwards by the infiltrating water. Thus, the problem is mainly shifted to the groundwater. Could you discuss the overall success of the presented remediation experiment against this background?

Reply: See reply to second part of comment p.15 in lines 89-95 of this document Technical corrections

Comment: References: Bauterse et al (2000) and Stumpp et al. (2009) are not mentioned in the text Reply: Will be corrected it in the revised manuscript

Comment: Fig. 3: the legend is missing

Reply: Will be corrected in the revised manuscript

Comment: Fig. 4/5: explain the meaning of the red arrows.

Reply: The red arrows emphasize the variation in perchlorate concentration in time. In Figure 4 it describe perchlorate reduction in the upper 13 m while in figure 5 the arrow emphasize the increase in perchlorate concentration with time in the deeper section of the vadose zone. Elaboration on the meaning of the arrows will be added to the figure captions.

Please also note the supplement to this comment:
http://www.hydrol-earth-syst-sci-discuss.net/hess-2016-663/hess-2016-663-AC1-supplement.pdf

———————————————————

[Figure]

**Fig. 1.** igure 1. Ethanol VS DOC in all water samples where both were measured

---

## Author Response (AR1)

Dear Editor

Follows our reply to Editor Decision on the manuscript "Transport and degradation of perchlorate in deep vadose zone: implications from direct observations during bioremediation treatment" from May 28, 2017. In addition the revised manuscript includes also the corrections that follow our reply to reviewer comments from March 19, 2017 (enclosed below). All changes in the revised manuscript are highlighted. Note that the tables and figures were moved to the end of the revised manuscript.

Reply to Editor Decision (May 28 2017)

Comment 1: One of the main comments in Editor decision letter (from March 28) regards *"representativeness of single point observations in different depths… for… the entire vadose profile…is not at all straightforward…"* and may be "*misleading"*, since it "*implies that a plot of concentrations at different depths into a single profile … suggests that these observation points are connected by a vertical flow paths…"*.

Reply: We are well aware the problem. We deal with graphical presentation of VMS data since we first introduced the VMS concept in 2003. We are aware to the fact that people are used seeing profile data sets as vertical profiles. Nevertheless, the VMS produce data from multiple points across the entire vadose zone. The points are not under a vertical profile but are very close to each other ($35^o$ means 70 cm horizontal shift for each 100 cm). We have tried many other graphic alternatives to show depth variation across the unsaturated zone. We did not find yet a better way to present the data in a clear and simple display which will not be a burden to the reader to understand the time variation in concentration with depth. We know that this kind of visualization is a necessary compromise. Accordingly, we used this method in few other publications (for example see Dahan et al 2014 HESS). In any event, the data points from the vadose zone are connected with a line only to emphasize the time variation in concentration across the unsaturated zone. Nevertheless, in order to prevent potential confusion the first paragraph of the Result and discussion chapter (Lines 225-233 in the revised manuscript) explicitly describe the following:

"*All of the data obtained by the VMS are presented here as variations in measured parameters with depth, as commonly done to describe depth profiles. However, to ensure measurements under undisturbed vertical profiles, the VMS was installed in a slanted orientation (Fig. 2 and supplementary material). Thus,* **each monitoring unit faces an undisturbed profile that is shifted horizontally and vertically from the other units. Accordingly, although the data are presented as depth profiles, they should be regarded as individu**al **points distributed across the 3D space of the vadose zone** (Dahan et al., 2007; Rimon et al., 2011a).

In addition the figure captions of all profile figures was revised to emphasize that the line is not meant for spatial interpolation but showing the profiles as time series. Lines 598,605, 616 and 621 in the revised manuscript "*Note that data points are aligned in a slanted orientation and interpolated as time intervals"*. Also interpolation lines appearance in Fig. 4, Fig. 5. Fig. 7 and Fig. 8 were changed to a thinner dashed line to weaken its appearance.

Comment 2: Further on in the Editor Decision letter the editor climes that the "so called" interpolation lines *"would be correct a) in case the perchlorate concentrations are homogenous in a given depth, and b) the flow and transport process during irrigation would be a rather uniform"*.

Reply: This is not something that we claim or believe in any way. Neither homogeneity nor uniformity flow pattern, could not be claimed even if measurement would have been taken under a vertical profile. Spatial variations in water content values, as measured by the FTDR sensors, are primarily related to the grain size distribution and water retention properties of the sediment. Clearly, sediment with different texture will have different volumetric water content under the same water potential. This is why each sensor sees a different level of water content. Nevertheless, in our reply to reviewer #2 (line 97; comment on line 287) we showed that the wetting front propagation velocity across the upper 10 m of the unsaturated zone is relatively uniform even though all sensors are not under the same vertical profile and even though the water amount that was used for each infiltration event was significantly different. Very similar observations were previously reported at Dahan et al 2007, 2008 and Rimon et al 2007.

Comment 3: *"By the way, please add a legend to this Figure relation the color codes to the depths and position of the FTDR".*

Reply: See reply to comments made by reviewer #2 (75-94 in reply to reviewer 2). Depth marks and notations to the infiltration events were added to the figure 3.

Comment 4: *"the representativeness of the VMS data for the entire domain needs to be carefully addressed … and… using a model is certainly way to address this issue.*

Reply: We fully respect this approach and used it in many of our publication where the VMS was used (see Dahan et al., 2007; Turkeltaub et al., 2014 VZJ; Turkeltaub et al., 2015a WRR; Turkeltaub et al., 2015b JHudrol; and Turkeltaub et al., 2016 HESS). Nevertheless, in our reply to reviewer 2 general comment , we presented  a whole discussion on the meaning on the model vs. data. When a model is stronger than the data and when the model do not provide any addition insight. In our case we show that the data is sufficient and eliminate the need for model. For example, a 1D flow and transport model was developed for the unsaturated zone at the site. The model was calibrated and validated on the basis of the data on variations in water content and bromide concentration that is presented in this manuscript. The model was developed as a 1D though and the data is from multiple parallel vertical profiles which are the outcome of the slanted installation. Nevertheless, the model results shows a relatively good fit to the data although it is absolutely a relatively large scale compares common tracers experiments in the vadose zone. Accordingly we believe that the manuscript will not benefit much if we add the model. It will definitely make it longer but not better. Similar examples were also demonstrated in some of the cited publications Turkeltaub 2015, 2016.

[Figure]

Figure xx. Transport model calibration.

*Figure 1. Measured Vs. modeled water content (left) and Bromide (right) variation at various depth during infiltration experiments.*

**References**

Dahan, O., Shani, Y., Enzel, Y., Yechieli, Y. and Yakirevich, A.: Direct measurements of floodwater infiltration into shallow alluvial aquifers, J. Hydrol., 344(3–4), 157–170, 2007.

Rimon, Y., Dahan, O., Nativ, R. and Geyer, S.: Water percolation through the deep vadose zone and groundwater recharge: Preliminary results based on a new vadose zone monitoring system, Water Resour. Res., 43(5), 1–12, doi:10.1029/2006WR004855, 2007.

Rimon, Y., Nativ, R. and Dahan, O.: Physical and Chemical Evidence for Pore-Scale Dual-Domain Flow in the Vadose Zone, Vadose Zo. J., 10(1), 322, doi:10.2136/vzj2009.0113, 2011a.

Turkeltaub, T., Dahan, O. and Kurtzman, D.: Investigation of Groundwater Recharge under Agricultural Fields Using Transient Deep Vadose Zone Data, Vadose Zo. J., 13(4), doi:10.2136/vzj2013.10.0176, 2014.

Turkeltaub, T., Kurtzman, D., Bel, G. and Dahan, O.: Examination of groundwater recharge with a calibrated/validated flow model of the deep vadose zone, J. Hydrol., 522, 618–627, doi:10.1016/j.jhydrol.2015.01.026, 2015a.

Turkeltaub, T., Kurtzman, D., Russak, E. and Dahan, O.: Water Resources Research, Water Resour Res., 4840–4847, doi:10.1002/2015WR017273.Received, 2015b.

Turkeltaub, T., Kurtzman, D. and Dahan, O.: Real-time monitoring of nitrate transport in the deep vadose zone under a crop field – implications for groundwater protection, Hydrol. Earth Syst. Sci., 20(8), 3099–3108, doi:10.5194/hess-20-3099-2016, 2016.

Reply to reviewer #1 comments on the manuscript:

**Transport and degradation of perchlorate in deep vadose zone: implications from direct**
**observations during bioremediation treatment**

We thank the reviewer for his constructive review and we address all of his comments in the
reply below. We would like to state that we are specifically encouraged by his statement "*The*
*presented topic is of relevance for many sites worldwide, polluted with different chemicals which*
*can be deactivated by microbial processes. The specific challenge of this approach was the*
*location of the pollution within a deep vadose zone with complicated water flow conditions*". In
our view this is the main essence of this manuscript.

General comments

*Comment: Some context would be easier to understand if the order of subsections would be*
*rearranged. For example: Section 4.3 explains why the different treatments for the experiments*
*were chosen, because the infiltration depth was not sufficient in the beginning and the*
*concentration of ethanol was too low during the first experiment. It would be good to have this*
*information already in the beginning before the results of perchlorate transformation are shown*
*and discussed. The same is true for the presentation of bromide tracer behavior (in the*
*beginning of section 4.4) which again explains the experimental setup.*

Reply: We accept the comment. On top of the detailed description of the experimental setup in
chapter "3.3 Infiltration experiment" A section describing the overall structure of all three
experiments was added to the beginning of the result chapter. It presents the rationale behind
all experiments and gives an overview of the measurements before detailed description of the
various components (lines 234 – 251 in the revised manuscript).

Specific comments

*Comment: p. 5, l. 111: You state that perchlorate is slowly leached into the groundwater. Can*
*you describe the behavior of this pollutant in the saturated zone? Is it reduced or only*
*transported by groundwater flows?*

Reply: Perchlorate is well known to be fairly stable in groundwater. Its natural degradation is
very limited and it is highly mobile. This has been presented in several publications (See for
example a review paper by Bardiya et al. 2011, a chapter in a book Coates JD, Gu B. 2006, and
perchlorate mobilization in this particular site Gal et al. 2009 (all of which are cited in this
manuscript). The possibility of perchlorate bio-reduction is depend in the groundwater redox
conditions.  We had reported  in the past that groundwater is aerobic and thus natural
degradation of perchlorate is not expected (Bernstein et al., 2010). Since our manuscript focus
on the unsaturated zone where the hydro-chemical and biological conditions are substantially
different from those occurring in groundwater we rather to focus on the unsaturated zone and
not elaborate on the saturated part beyond the limited citations in the introduction chapter.

(Bernstein, A., Adar, E., Ronen, Z., Lowag, H., Stichler, W., & Meckenstock, R. U. (2010).
Quantifying RDX biodegradation in groundwater using δ 15 N isotope analysis. Journal of
contaminant hydrology, 111(1), 25-35.)

*Comment: p. 6, l. 147: What is the effect of these climatic conditions? Is the perchlorate only*
*transported during the winter season and probably rises again during summer due to capillary*
*action?*

Reply: The vadose zone is very thick (~40 m) and mostly sandy. As such, the capillary action
relevant only to the bottom ~1 m of the unsaturated zone. The experimental area has been
covered with a sealing polyethylene liner to prevent air penetration and to promote anoxic
conditions in the vadose zone. As such the only source of water to the subsurface in this period,
is the water introduced to the soil with the drip irrigation system under the surface cover.
Accordingly the consequence of rain water infiltration is excluded. In addition in such thick
vadose zone even seasonal temperature fluctuations are limited to the upper 2 m (Rimon et al.
2011b, cited in the manuscript). As such we believe that the climate has only limited impact on
the conditions in the subsurface.

*Comment: p. 11, l. 229: Please explain why no tracer was used in the second and third*
*application.*

Replay: A single slug of tracer was used in in the beginning of the first experiment. It was
designed to enable tracing of the wetting front that was introduced to the subsurface during the
experiment. Application the tracer in the following experiment would have result in smearing the
identity of the front and masking our capability to trace the moving water. In well-defined
medium such as column experiment it is possible to differ between tracers applied in different
stages. Yet we tend to believe that in natural heterogeneous system where water flow may be
subjected to multi flow trajectories that may be activated and deactivated according to the
hydraulic condition (see Dahan et al. 2009), application of the tracer in the following
experiments would be a disadvantage.

*Comment: p. 12, l. 272 Can you exclude lateral fluxes of seepage water?*

Reply: We cannot absolutely exclude local limited of lateral fluxes. Nevertheless, creation of
lateral flow in the unsaturated zone require, by definition, generation of saturated conditions that
will create positive pressure which could overcame gravitational drainage. Up to date the
vadose zone monitoring system has been installed in dozens of sites with different geological
and hydrological conditions (See for example Dahan et al., 2007, Dahan et al. 2008, Rimon et al
2007, 2011a, 2011b, Amiaz et al 2012 and others). In none of these sites we found evidences
for creation of saturation conditions and thus creation of lateral flow in the vadose zone, even
though, some of the sites were under flooded conditions of high water head (Dahan et al 2007,
2008), some with geological formations which are composed of clay interbeds that could
potential create some kind of hydrological barrier and lateral flow. Since we did not find any
indication for lateral flow in any of the other studies where water flow in the vadose zone was
monitored, we believe that in this particular site lateral flow, if any, was very limited.  In this discussion we ignored lateral small scale capillary flow and lateral flow in purged aquifers. Both
are not relevant to this site.

*Comment: p. 15, l. 326: Is the described successful reduction of perchlorate concentration the*
*result of transport or reduction processes? Would it be a success if perchlorate is mainly*
*transported by seepage water into deeper parts of the soil?*

*Comment: p. 16, l. 333: You mention mixed trends for both transformation and mobilization*
*processes. Could you explain this conclusion more in detail?*

Reply to the two comment above (p.15 and p.16): This comment emphasize the greatest
challenge we faced in this project. Can we absolutely state that the reduction in perchlorate
concentration that we have observed in the upper parts of the unsaturated zone are the result of
bio-degradation or simple down leaching with the percolating water. Moreover, we have to
investigate this question in light of the fact that the concentration of perchlorate in some deep
section only increased during the infiltration experiments (Figure 4 and 5 in the manuscript).
Throughout the paper we have discussed the potential degradation versus leaching from
different perspectives. In section 4.3 we have analyzed the potential degradation of perchlorate
to the availability of electron donor. Obviously, under the absence of available electron donor;
no perchlorate degradation will take place. Though we managed to introduce electron donor into
the vadose zone it was limited to the top 13 m. Only at this section, we had found some bio-
reduction in perchlorate. In the rest of the profile we found no increase in available electron
donor and in fact we also found no reduction perchlorate concentration. On the contrary, in
some places, the concentration only increased which is an obvious indication to perchlorate
mobilization with the percolating water. Further down in the manuscript in section 4.4 we
discussed the potential degradation of perchlorate versus its transport through a comparison of
the ethanol migration, which was consumed, versus the tracer, Br. Here we also compared the
reduction in perchlorate with the variations in concentration of its final degradation product
chloride,  across the unsaturated zone and found a pronounced increase in Cl/Perchlorate only
in the zones where we found available electron donor. All of these indicators provided hints to
the question on the degradation vs leaching.

In the second part of the first comment the reviewer ask if "it be a success if perchlorate is
mainly transported by seepage water into deeper parts of the soil". This is a very important
question that is the subject of several studies we are conducting now (See Avishai et al 2016.
Journal of Hazardous Materials). Since we found that inducing "efficient" degrading conditions in
the deep vadose zone is limited and we suggested that perchlorate mobilization in the
unsaturated zone is very high we are testing the possibility to leach the pollution down to the
groundwater where it can be retrieved back for treatment on land surface.

*Comment: p. 17, l. 350: Probably the relation between ethanol concentration and DOC could be*
*shown by means of a figure and a regression curve?*

Reply: As mentioned in p.13 lines 304-306 (in the revised manuscript), we found high
correlation between ethanol and DOC.  Even though  ethanol is mineralized by perchlorate reducing bacteria, it may degraded first to acetate that also serve as energy for the degrading
bacteria thus, DOC provide better picture on the availability of electron donor in the soil pore
water. Since it is all presented in the manuscript text, we believe that adding this information in a
figure is somewhat not necessary.  The figure below display ethanol vs DOC in all water
samples were both ethanol and DOC were measured (we do not think that adding this chart to
the manuscript is necessary).

[Figure]

Figure 1. Ethanol VS DOC in all water samples where both were measured

*Comment: p. 21, fig. 8: Is the red graph an average for data of the period 1/3-11/4 2015 (1.5*
*months)?*

Reply: The red graph is a combination of data obtained from two consequent sampling data.
Due to a technical problem that was resulted in luck of samples from one of the dates it was
necessary to integrate data from these two consequent dates. Nevertheless, the figure legend
was slightly modified to emphasize that the dates are 1/3 & 11/4, 2011

*Comment: p. 22, l. 459: You end up with the conclusion that the entire column of perchlorate*
*was pushed downwards by the infiltrating water. Thus, the problem is mainly shifted to the*
*groundwater. Could you discuss the overall success of the presented remediation experiment*
*against this background?*

Reply: See reply to second part of comment p.15 in lines 103-109 of this document

Technical corrections

*Comment: References: Bauterse et al (2000) and Stumpp et al. (2009) are not mentioned in the*
*text*

Reply: Comment accepted and the manuscript was revised (lines 58, 60 and 63 in the revised
manuscript)

*Comment: Fig. 3: the legend is missing*

Reply: Figure 3 was revised accordingly

*Comment: Fig. 4/5: explain the meaning of the red arrows.*

Reply: The red arrows emphasize the variation in perchlorate concentration in time. In Figure 4
it describe perchlorate reduction in the upper 13 m while in figure 5 the arrow emphasize the
increase in perchlorate concentration with time in the deeper section of the vadose zone.
Elaboration on the meaning of the arrows was added to the figure captions (lines 597 and 604 in
the revised manuscript).

Reply to reviewer # 2 comments on the manuscript:

**Transport and degradation of perchlorate in deep vadose zone: implications from direct**
**observations during bioremediation treatment**

We would like to express our great appreciation to the reviewer comments and believe that we
addressed all questions and comments raised in this review.

General comments

*Comment: The major concerns are: i) the absence of any quantitative modelling of the water*
*transport and/or the perchlorate pollution plume during the infiltration experiment; ii) the*
*absence of any uncertainty assessment. Hypothesis related to the fate of the perchlorate plume*
*are indeed subjected to the hypothesis of mass conservation and representativity of the singular*
*sampling. These strong hypotheses can only be considered acceptable in the present case if*
*the experimental results are compared with some quantitative modelling that are built on mass*
*conservation principles ( using e.g. a numerical water and solute transport, or NAPL/DNAPL*
*transport model). As long as this numerical modelling is not added to the paper, the results*
*remain too much speculative*

Reply: The reviewer concerns regarding absence of a quantitative model on water flow and
solute transport may be addressed in this manuscript. In fact a calibrated model that is based on
the measured hydraulic and chemical properties of the vadose zone has been constructed and
can be add to the manuscript. Nevertheless, during the manuscript preparation we have
decided to omit the model chapter from this manuscript. The reason is simply because we have
found that the strength of this manuscript lay in the long-term continuous data obtained from the
entire flow domain and not from the model which obviously was based on the measured
parameters. Moreover, we have found that the model did not add any valuable information that
could not be observed directly from the measured data. The value of hypothesis based on a
model vs hypothesis base on observation is a fundamental argument that requires a critical
discussion before implementation.

Modeling by definition aims at extending knowledge from limited data set that may be obtained
from small scale point measurements or information from the domain boundaries into larger
scales or zones where the knowledge is limited. For example, vadose zone modeling often uses
information from the domain boundaries at or near land surface, to understand processes taking
place within the unsaturated zone where data on the dynamics of water flow and solute
transport is limited. Nevertheless, the model inherently bear substantial amount of basic
assumption and therefore "quantitative modeling" is by definition speculative. However, in
absence of quantitative observations on the flow dynamics within the domain, as often found in
vadose zone studies, the model is the only practical tools for processes quantification.
Nevertheless, whenever the hydraulic or chemical characteristics within the domain can be
measured continuously  and provided direct indication to the dynamics of flow and transport, as
demonstrated in our manuscript, then modeling is not the "sol and only" mean for quantitative analysis. It is obvious that monitoring and measurements in the unsaturated zone, sophisticated
as can be, are also limited in their capability to describe the flow and transport processes
(technology and method dependency). Therefore, the implications from both, the model
approach and the monitoring approach are, to some extent, speculative and not presenting the
"truth and nothing but the truth". In this manuscript we used for direct and continuous
measurements of hydraulic and chemical characteristics of unsaturated zone to quantification
the dynamics of water flow and solute transport within the entire domain. Nevertheless, general
results from a relevant model are presented in reply to comment 4 in Editor Decision letter.

Specific comments

*Comment: Line 103. Study site. Can the origin of perchlorate in the study site be identified?*

Reply: The site is a former waste pond of an ammonium perchlorate factory. The origin of the
perchlorate in the soil is well defined, as described in details in Gal et al. 2008, 2009.

*Comment: Line 121. Heterogeneity in sedimentary vadose zone formations is omnipresent.*
*Hence, how reliable is the single borehole to assess the lithology of the study site. Is the*
*information of the borehole consistent with information obtained from the boreholes in the*
*vicinity of the sampling point?*

Reply: In this manuscript we present the lithology and concentration as measured in a borehole
that was drilled for this project in the center of the experiment site (30X10 m). Nevertheless,
several other boreholes were drilled in this site and a general agreement in both lithology and
concentration profiles were found (Gal et al., 2008, 2009). This has been expressed in the
manuscript p. 5 line 11-114 in the revised manuscript.

*Comment: Line 152. The high suspected correlation between chloride and perchlorate*
*concentrations demonstrates that there is some natural attenuation. This is in contrast with the*
*statement in the literature review (line 86).*

Reply: The limited natural attenuation of perchlorate in the site was reported extensively in Gal
et al 2008, 2009. Nevertheless we do not understand how chloride/perchlorate correlation
demonstrates natural attenuation. On the contrary, perchlorate reduction should have been
resulted in increased chloride/perchlorate ration as demonstrated in figure 8. It is important to
note that chloride was present in the soil (from the waste pond) as described previously. It is
not possible to say that the chloride originated from perchlorate reduction Gal et al 2008, 2009.

*Comment: Line 198. Explain more in detail how ethanol can eliminate increased salinity.*

Reply: One of the most common electron donor used for perchlorate bio-degradation is Sodium
acetate. Therefore, application of large amounts of sodium-acetate may end-up in salinization
and potentially sodification of the vadose zone. Using natural substrate will not introduce more
ions like sodium into the soil.

*Comment: Line 214. Specify for each infiltration pulse how much time was needed to apply the*
*water/tracer/ethanol (hence the application rates). Also, add an estimate of the saturated*
*hydraulic conductivity of the different layers to demonstrate that the infiltration rates stayed*
*sufficiently below the ponding infiltration rate.*

Reply: Infiltration pulses were applied through a drip irrigation system with a constant drip rate
of 2.2 l/h and in a distribution of 0.3X0.3 m (stated in line 157 in the revised manuscript).
Accordingly, the application rate is 0.024 m/h, which is far below the soil Ks which is ~1 m/h
(loamy sand). As such, the application time of each phase is derived directly from the volume
divided by the discharge rate. All of which appears in chapter 3.3 Infiltration experiment and
table 2 (For clarification see lines 155-160 in the revised manuscript). No ponding conditions
were observed on surface and the sediment water content in the unsaturated zone remain
below saturation. Due to a technical mistake during submission the water content hydrographs
(figure 3 in original manuscript) was submitted without the legend and depth specification.
Figure 1 below includes this missing information. Note that in any case the water application
time in all infiltration events was in the scale of hours (7, 14, and 42 h) compare with the
variation in the vadose zone water content, as presented in figure 3, is in time scale of months.

[Figure]

Figure 1 (figure 3 in the revised manuscript).  Temporal variations in sediment water content in
the top 13 m of the vadose zone during the infiltration experiments. Dates are given as
day/month/year.

*Comment: Line 250. Significant at which statistical level?*

Reply: see reply to comment p 17 of reviewer 1

*Comment: Line 287. Specify exactly how the wetting front velocities are determined. We are*
*definitely in strong transient flow conditions. Hence the wetting front velocities will vary*
*dynamically in time.*

Reply: it is obvious that an infiltration event creates field of velocities that dynamically vary in
space and time. Yet, (as stated in lines 258-260 in the revised manuscript ), the wetting front
propagation velocity, which reflect the natural gravitational drainage across the unsaturated
zone, is calculated from the wetting sequence with respect to the infiltration events on land
surface. The figure below describes the wetting sequence with depth at the 3 infiltration
experiments. It present the time from initiation of the infiltration event to the measured
increase in water content as shown in figure 2. In addition, Table 1 in this document
describes the calculated velocities to the various depths in all three experiments.

[Figure]

Figure 2. Wetting front propagation in the upper part of the vadose zone during all three
infiltration experiments, represented by the time of first measured increase in water content V.S.
depth.

**Table 1. Velocity calculation for wetting front propagation**

|  | first infiltration experiment | | second infiltration experiment | | third infiltration experiment | |
|---|---|---|---|---|---|---|
| Depth (m) | arrival time (hr) | velocity (m/hr) | arrival time (hr) | velocity (m/hr) | arrival time (hr) | velocity (m/hr) |
| 0.5 | N/D | N/D | 5 | 0.10 | 7 | 0.07 |
| 2.6 | 20 | 0.13 | 13 | 0.20 | 16 | 0.16 |
| 5.5 | 28 | 0.20 | 25 | 0.22 | 25 | 0.22 |
| 8.4 | 40 | 0.21 | 37 | 0.23 | 33 | 0.25 |
| 11.2 | N/D | N/D | N/D | N/D | 142 | 0.08 |

*Comment: Line 290. Be more rigorous and more specific with respect to 'flow velocities'. How are these "flow velocities" defined in a heterogeneous and time dynamic flow system? (Cf.a major concern on the need to confront such statements with those from a quantitative numerical model).*

Reply: Direct calculating of wetting front propagation velocity from the temporal variation in the vadose zone water content is a basic technique which has been described in numerous publications (Dahan et al 2007, 2008, 2009, Rimon et al 2007, 2011, all of which are cited in the manuscript). It has been further used to calibrate flow and transport models in the unsaturated zone (Turkeltaub 2014, 2015a, 2015b, 2016). As stated above, whenever high resolution hydraulic data may be obtained from the unsaturated zone then modeling is not the "only" quantitative tool. And direct measurement of flow velocities is achievable.

*Comment: Line 302. Legend incomplete. What are the different coloured curves? Where are the results of the 11 sampling units? Quid results of the control units in the top layer (0,5 and 1.3 m depths)?*

Reply: The comment is absolutely right, and we are regret for this technical mistake (see figure 1 here). The correct figure 3 with all necessary legend information was added to the revised manuscript.

*Comment: Line 302. Explain more in detail the observed curves. E.g. what happens with the TDR probe at the top (I suppose) during the third infiltration event? The drainage curve looks completely different. So what happened?*

Reply: We agree that it was hard to understand the wetting and drainage cycles without the legend and further explanation of the velocity calculation. We hope with our reply to previous three comments the subject is now clearer.

*Comment: Line 356. This statement can't be supported. This can only be concluded if mass*
*conservation is checked. You can have lateral flow dissipation in such system. Only, a*
*comparison of the results with the results of a numerical mass conservative model can support*
*such conclusions.*

Reply: We can hardly agree with the reviewer comment that "Only, a comparison of the results
with the results of a numerical mass conservative model can support such conclusions". In this
section (Lines 309-314 in the revised manuscript) we describe how continuous measurement of
ethanol concentration across the profile dropped too practically zero.  What is it if not a direct
mass conservation check; which show that the entire mass of ethanol had consumed as a result
of microbial activity? No model can give higher degree of confidence in such mass balance.
Especially, when it is compared with the transport of a conservative tracer such as Br. We have
dedicated a special chapter (4.4 transport and degradation) which deal with mas conservation of
degradable and non-degradable substance during infiltration experiment.

*Comment: Line 400-402. Show this in an explicit way.*

Reply: Here again we present the dynamic variation in concentration of degradable (ethanol)
and no degradable (Br) substance transported together in the unsaturated zone. We show how
the mass of Br is conserved while the mass of ethanol is reduced in an environment that is by
definition biologically active. It is presented as time series of the ethanol (figure 6 in the
manuscript) along time series of Br (presented as profile variations in figure 7). Accordingly we
do not understand what is the meaning of more explicit way.

*Comment: Line 426. Confusing legend. 1/3 -11/4 2011. Specify which data at which date*
*exactly.*

Reply: Due to technical analytical problem we had to combine data from two consequent dates
1 March 2011 and 11 April 2011, which represent the ending period. Nevertheless, we revised
the legend to improve its clarity (1/3 & 11/4, 2011 )

*Comment: Line 451. There are other studies showing that the clay layers will have considerable*
*impact on the vadose zone dispersion (See e.g. Javaux M. and M. Vanclooster, 2004. In situ*
*long-term chloride transport through a layered, non-saturated subsoil.1. Data set, interpolation*
*methodology and results. Vadose zone journal 3 : 1331-1339.).*

Reply: We fully agree with the reviewer comment that a clay layer in the unsaturated zone may
impact the dispersion. In fact this is something that we also found in our studies on water
infiltration in layered vadose zone. Nevertheless, our statement refers to the infiltration capacity,
in terms of flow velocity and fluxes.  Several different and independent studies showed that the
presence of the clay layer in the unsaturated zone do not limit the flow velocity (Dahan et al
2009, Rimon et al 2007, 2011, Baram et al 2012, Turkeltaub 2015). A clarification sentence was
added to the manuscript (lines 390-391 in the revised manuscript).

*Comment: Line 461. This has not been shown in the paper.*

Reply: The reviewer statement that the sentence "It seems that the entire column of
perchlorate mass was pushed down by the percolating water toward the water table,
which also resulted in an increased concentration of perchlorate in the observation well,
which was located under the infiltration zone." has not been shown in the paper is not
clear. Figure 5 presents variation in perchlorate concentration profile during the
infiltration experiment. It exhibit increased concentration of perchlorate in zones
underlying layers of higher concentration as a response to water infiltration. This is a
unequivocal indication to solute displacement.

**References**

Baram, S., Arnon, S., Ronen, Z., Kurtzman, D. and Dahan, O.: Infiltration Mechanism Controls
Nitrification and Denitrification Processes under Dairy Waste Lagoon, J. Environ. Qual., 41(5),
1623–1632, doi:10.2134/jeq2012.0015, 2012b.

Baram, S., Kurtzman, D. and Dahan, O.: Water percolation through a clayey vadose zone, J.
Hydrol., 424–425, 165–171, doi:10.1016/j.jhydrol.2011.12.040, 2012c.

Dahan, Talby, R., Yechieli, Y., Adar, E., Lazarovitch, N. and Enzel, Y.: In Situ Monitoring of
Water Percolation and Solute Transport Using a Vadose Zone Monitoring System, Vadose Zo.
J., 8(4), 916–925, 2009.

Dahan, O., Shani, Y., Enzel, Y., Yechieli, Y. and Yakirevich, A.: Direct measurements of
floodwater infiltration into shallow alluvial aquifers, J. Hydrol., 344(3–4), 157–170, 2007.

Dahan, O., Tatarsky, B., Enzel, Y., Kulls, C., Seely, M. and Benito, G.: Dynamics of flood water
infiltration and ground water recharge in hyperarid desert, Ground Water, 46(3), 450–461,
doi:10.1111/j.1745-6584.2007.00414.x, 2008.

Dahan, O., Babad, A., Lazarovitch, N., Russak, E. E. and Kurtzman, D.: Nitrate leaching from
intensive organic farms to groundwater, Hydrol. Earth Syst. Sci. Discuss., 10(7), 9915–9941,
doi:10.5194/hessd-10-9915-2013, 2014.

Gal, H., Ronen, Z., Weisbrod, N., Dahan, O. and Nativ, R.: Perchlorate biodegradation in
contaminated soils and the deep unsaturated zone, Soil Biol. Biochem., 40(7), 1751–1757,
doi:10.1016/j.soilbio.2008.02.015, 2008.

Gal, H., Weisbrod, N., Dahan, O., Ronen, Z. and Nativ, R.: Perchlorate accumulation and
migration in the deep vadose zone in a semiarid region, J. Hydrol., 378(1–2), 142–149,
doi:10.1016/j.jhydrol.2009.09.018, 2009.

Turkeltaub, T., Dahan, O. and Kurtzman, D.: Investigation of Groundwater Recharge under
Agricultural Fields Using Transient Deep Vadose Zone Data, Vadose Zo. J., 13(4),
doi:10.2136/vzj2013.10.0176, 2014.

Turkeltaub, T., Kurtzman, D., Bel, G. and Dahan, O.: Examination of groundwater recharge with
a calibrated/validated flow model of the deep vadose zone, J. Hydrol., 522, 618–627,
doi:10.1016/j.jhydrol.2015.01.026, 2015a.

Turkeltaub, T., Kurtzman, D., Russak, E. and Dahan, O.: Water Resources Research, Water
Resour Res., 4840–4847, doi:10.1002/2015WR017273.Received, 2015b.

Turkeltaub, T., Kurtzman, D. and Dahan, O.: Real-time monitoring of nitrate transport in the
deep vadose zone under a crop field – implications for groundwater protection, Hydrol. Earth
Syst. Sci., 20(8), 3099–3108, doi:10.5194/hess-20-3099-2016, 2016.

---

## Author Response (AR2)

Reply to reviewer comments,

Prior to our detailed reply for the comments we which to express our great appreciation to the editor Erwin Zehe, the reviewers Marnik Vanclooster and anonymous reviewer #1. We had found the discussion during the review process very fruitful and encouraging. It improved this paper and helped us to get additional perspectives for the coming publications. All comments were accepted and addressed in both the supplementary material and revised manuscript (highlighted in revised manuscript). Also following the revision process we added Lior Avishai as an author to this manuscript.

Follows our reply to the comment. In addition to the reply to

Reply to Report # 1

Comment: ... a comparision with model results would be of importance for illustrating the representativeness of the measured values. The results of such a model application are shown in the authors' reply and should be added (along with the description) to the supplemental material of this paper. The same is true e.g. for the good correlation between ethanol and DOC. Further, with regard to the discussion of effectiveness of degrading or just transporting the contaminant to deeper parts of the soil …it would be good to have this discussion within the conclusions part of the paper….

Reply: We accept all comments an revised the manuscript and the supplementary material to account for: (a) comparison of the data with a 1D flow and transport model (lines 104 -119) in the supplementary material, (b) correlation between DOC and ethanol concentration in water samples obtained from the vadose zone (lines 125-132 in the supplementary material and in lines 305-315 in the revised manuscript), and (c) revising the conclusion chapter in the manuscript to elaborate on perchlorate degradation VS migration processes (lines 412 –424 in revised manuscript).

Reply to Report # 2, Marnik Vancluster

Comment: Line 165: It does not eliminate, but reduce salinity.

Reply: Accepted. The sentence was revised (Line 165 in the revised manuscript).

Comment: Line 306: DOC (dissolved organic carbon) should by definition be expressed in concentration units. So 2g DOC sounds odd.

Analysis of ethanol and DOC in the water samples from the vadose zone throughout the experiment revealed high correlation between the two. Theoretically, one gram per liter of ethanol is equal to 0.52 gram per liter of soluble carbon. However, in the site, the dissolved carbon compose of ethanol its oxidation products (such as acetate) as well as well other solubale microbial metabolites that can also serve as electron donors . Thus, DOC provide a better knowledge on the availability of electron donor in the soil pore water (lines 305-312 in the revised manuscript).

Comment: Line 326: What is the 50 % referring to? Mass percentage? Volume percentages? I suggest using concentration units.

Reply: Accepted. The definition of 50% volume percentage was added to the manuscript (line 335 in the revised manuscript).

Comment: Revision report: Comment: Line 287. The authors gave a detailed reply on how velocities were calculated in the profile, based on the wetting front appearances at different position in the profile. This information is crucial to understand the hydrodynamics of the system. The calculation procedure should therefore be included. I suggest to include the calculation procedure as supplementary material to the manuscript.

Reply: We accept the comment and add the main features from our reply on velocity calculation to supplementary material (lines 86-102) and relevant notation in the manuscript (lines 263-265  in the revised manuscript ).